# SPOP Deregulation Improves the Radiation Response of Prostate Cancer Models by Impairing DNA Damage Repair

**DOI:** 10.3390/cancers12061462

**Published:** 2020-06-04

**Authors:** Rihan El Bezawy, Martina Tripari, Stefano Percio, Alessandro Cicchetti, Monica Tortoreto, Claudio Stucchi, Stella Tinelli, Valentina Zuco, Valentina Doldi, Paolo Gandellini, Riccardo Valdagni, Nadia Zaffaroni

**Affiliations:** 1Molecular Pharmacology Unit, Department of Applied Research and Technological Development, Fondazione IRCCS Istituto Nazionale dei Tumori, Via Amadeo 42, 20133 Milan, Italy; Rihan.elbezawy@istitutotumori.mi.it (R.E.B.); Stefano.percio@istitutotumori.mi.it (S.P.); Monica.tortoreto@istitutotumori.mi.it (M.T.); Stella.tinelli@istitutotumori.mi.it (S.T.); Valentina.zuco@istitutotumori.mi.it (V.Z.); Valentina.doldi@istitutotumori.mi.it (V.D.); 2Cellular and Molecular Physiology, Institute of Translational Medicine, University of Liverpool, Liverpool L69 3BX, UK; Martina.tripari@liverpool.ac.uk; 3Prostate Cancer Program, Fondazione IRCCS Istituto Nazionale dei Tumori, 20133 Milan, Italy; Alessandro.cicchetti@istitutotumori.mi.it (A.C.); Riccardo.valdagni@istitutotumori.mi.it (R.V.); 4Medical Physics Unit, Fondazione IRCCS Istituto Nazionale dei Tumori di Milano, 20133 Milan, Italy; Claudio.stucchi@istitutotumori.mi.it; 5Department of Biosciences, Università degli Studi di Milano, Via Celoria 26, 20122 Milan, Italy; Paolo.gandellini@unimi.it; 6Department of Oncology and Hemato-Oncology, Università degli Studi di Milano, 20122 Milan, Italy; 7Department of Radiation Oncology 1, Fondazione IRCCS Istituto Nazionale dei Tumori, 20133 Milan, Italy

**Keywords:** prostate cancer, radiosensitivity, SPOP, mutation

## Abstract

Speckle-type POZ (pox virus and zinc finger protein) protein (SPOP) is the most commonly mutated gene in prostate cancer (PCa). Recent evidence reports a role of SPOP in DNA damage response (DDR), indicating a possible impact of SPOP deregulation on PCa radiosensitivity. This study aimed to define the role of SPOP deregulation (by gene mutation or knockdown) as a radiosensitizing factor in PCa preclinical models. To express WT or mutant (Y87N, K129E and F133V) SPOP, DU145 and PC-3 cells were transfected with pMCV6 vectors. Sensitivity profiles were assessed using clonogenic assay and immunofluorescent staining of γH2AX and RAD51 foci. SCID xenografts were treated with 5 Gy single dose irradiation using an image-guided small animal irradiator. siRNA and miRNA mimics were used to silence SPOP or express the SPOP negative regulator miR-145, respectively. SPOP deregulation, by either gene mutation or knockdown, consistently enhanced the radiation response of PCa models by impairing DDR, as indicated by transcriptome analysis and functionally confirmed by decreased RAD51 foci. SPOP silencing also resulted in a significant downregulation of RAD51 and CHK1 expression, consistent with the impairment of homologous recombination. Our results indicate that SPOP deregulation plays a radiosensitizing role in PCa by impairing DDR via downregulation of RAD51 and CHK1.

## 1. Introduction

Speckle-type POZ (pox virus and zinc finger protein) protein (SPOP) is an adaptor for the E3 ubiquitin ligase Cullin 3 that selectively binds to and targets substrates for ubiquitination and proteasome degradation [1]. Genome-wide next generation sequencing studies have revealed that SPOP is frequently mutated in a number of cancer types such as prostate and endometrial [2,3], thus suggesting SPOP as a putative tumor suppressor. Counterintuitively, SPOP is overexpressed in other cancer types such as clear cell renal cell carcinoma, where it acts as a tumorigenic hub [4], indicating that the role of SPOP in cancer is likely tumor type dependent.

SPOP binds to its substrates via the N-terminal meprin and traf homology (MATH) domain [1], while it interacts with Cullin 3 through the bric-a-brac/tramtrack/broad complex (BTB) domain at the C-terminus [1]. SPOP mutations observed in human cancers are clustered in the MATH domain [2,3], suggesting that they may promote cancer via altering the stability of its substrates (Figure 1). SPOP mutations occur early in the natural history of prostate cancer (PCa) and are present in about 10% of both clinically localized and metastatic disease, thus representing the most common non-synonymous mutations in PCa [2,5,6]. SPOP mutations define a distinct molecular class of PCa, with a high frequency of characteristic genomic rearrangements, but they are mutually exclusive with rearrangement between TMPRSS2 and the ETS family of transcription factors (mainly TMPRSS2:ERG) [2,7].

A number of highly relevant substrates have been reported as deregulated by mutant SPOP in PCa cells, including the androgen receptor (AR) and its co-activators steroid-receptor co-activator-3 (SRC-3) and tripartite motif containing 24 (TRIM24) [8,9,10,11,12]. However, the relevance of these findings for prostate tumorigenesis is still unclear. Recent evidence obtained on a transgenic mouse with prostate-specific conditional expression of SPOP-F133V—the most frequent mutation observed in PCa—indicates that SPOP mutation alone is insufficient to drive tumorigenesis in the mouse prostate. Conversely, in the setting of phosphatase and tensin homolog (PTEN) loss, conditional expression of mutant SPOP in the prostate induced the appearance of high-grade prostatic intraepithelial neoplasia (PTEN heterozygous background) and invasive poorly differentiated carcinoma (PTEN homozygous background), suggesting that mutant SPOP is able to cooperate with PTEN to drive carcinogenesis [13]. In addition, using mouse prostate organoids, it was shown that, in PTEN WT background, SPOP-F133V not only promotes AR signaling but also induces SRC-3-mediated activation of PI3K/mTOR signaling, thus effectively uncoupling the normal negative feedback between these two pathways [13].

Still limited experimental data indicate a possible impact of mutant SPOP on treatment response. Results obtained in primary prostate cells from conditional SPOP-F133V transgenic mice, as well as in human normal prostate and PCa cell lines ectopically expressing SPOP-WT or SPOP-F133V, suggest that SPOP participates in the DNA damage response (DDR) and that SPOP mutation impairs homologous recombination (HR) [7], which is the main repair pathway of DNA double strand breaks (DSB). Consistently, it has been reported that SPOP loss impairs RAD51 foci formation in response to radiation-induced DNA DSBs [14]. Moreover, SPOP knockdown was found to lead to spontaneous replication stress and impaired recovery from replication fork stalling, due to the transcriptional suppression of RAD51 and CHK1, two main factors involved in HR [14]. In line with these findings, SPOP knockdown in human cervical carcinoma [15] and lung adenocarcinoma [16] cells resulted in impaired DDR and hypersensitivity to ionizing radiation.

Radiotherapy, alone or in association with androgen deprivation, is one of the main treatment options for organ-confined PCa [17]. Relevant advancements in treatment planning and delivery have significantly improved local tumor control. However, a considerable proportion of patients still experience recurrence due to resistance development [18]. The onset of radioresistance is a complex and still poorly understood phenomenon that hinges on the deregulation of a plethora of signaling pathways as a result of several genetic and epigenetic abnormalities [19].

In the present study, we aimed to define the role of cancer-specific SPOP mutations as radiosensitizing factors in preclinical models of PCa. The obtained findings were then confirmed in PCa cells in which SPOP expression was downregulated by siRNA or through the reconstitution of miR-145, which is a physiological negative regulator of the gene [20]. Results from the study, which represents the first attempt to directly compare the radiosensitizing effect of SPOP deregulation accomplished through different approaches, could contribute to tailoring therapy of individual PCa patients by delivering a genomic-adjusted radiation dose, instead of a uniform dose (i.e., one-size-fits-all) and to highlight the possibility of developing novel strategies of radiosensitization based on SPOP targeting.

## 2. Results

### 2.1. SPOP Mutation Sensitizes PCa Models to Ionizing Radiation

To investigate the potential of PCa-specific SPOP mutations as radiosensitizing factors in PCa, we transfected DU145 and PC-3 cell lines with pMCV6 expression vectors encoding for WT or mutant (Y87N, K129E and F133V) SPOP and assessed the effect on cell response to 2–8 Gy γ irradiation via clonogenic assay. Transfection with WT SPOP did not modify the plating efficiency of parental cells, while a reduced plating efficiency was consistently observed in cells transfected with mutant SPOP (Appendix A). Interestingly, although the overexpression of WT SPOP did not affect the radiosensitivity profile of either cell line, mutant SPOP ectopic expression enhanced radiosensitivity of both cell lines, independently of the type of mutation, as revealed by the reduction in their clonogenic cell survival compared to WT SPOP, which was statistically significant along the whole dose range (Figure 2A). Remarkably, the dose enhancement ratio (DER), here defined as the ratio of the radiation dose required to obtain a surviving fraction (SF) of 0.1 in WT SPOP cells to that required to obtain the same SF in mutant SPOP cells, was consistent for the different SPOP mutations (ranging from 1.31 to 1.43 in DU145 cells and from 1.22 to 1.31 in PC-3 cells), meaning that the radiosensitizing effect of mutant SPOP was independent of the type of mutation (Figure 2A).

To confirm the radiosensitizing effect observed in vitro also in an in vivo setting, SCID mice were subcutaneously transplanted with DU145 cells stably expressing F133V SPOP—the most frequent PCa-specific SPOP mutation [2]—to generate xenografts. Mice were exposed to 5 Gy single dose irradiation when tumors reached a volume of ~300 mm^3^ (24 days after cell inoculum). Interestingly, the ectopic expression of F133V SPOP increased the effect of radiation also in vivo, as shown by the statistically significant reduction in tumor growth upon irradiation compared to WT SPOP xenografts (Figure 2B).

Overall, these findings indicate that mutant SPOP is able to enhance PCa cell response to ionizing radiation both in vitro and in vivo.

### 2.2. SPOP Mutation Affects PCa Cell Radiation Response by Impairing Homologous Recombination

To get insight into the molecular mechanisms underpinning the radiosensitizing effect of mutant SPOP, a differential gene expression analysis comparing F133V SPOP with WT SPOP DU145 cells was carried out. Since among 12,457 assessed genes only 210 had a significant change (false discovery rate, FDR < 0.05) in their expression profile, we decided to evaluate modulation in pathways instead of single genes. To this purpose, we performed a pre-ranked gene set enrichment analysis (GSEA) on the Hallmark collection, which summarizes the main well-defined biological activities in a cell. By ranking gene expression according to the *t*-value obtained in the differential expression analysis, we obtained a list of significantly modulated gene sets, where the normalized enrichment score (NES) represents the magnitude of enrichment in F133V SPOP cells (Figure 3A). Notably, among the pathways enriched in F133V SPOP cells, we found an “androgen response” pathway, supporting the previously reported contribution of the mutant gene in up-regulating androgen receptor (AR) activity [13] (Figure 3B).

Consistent with the SPOP role in determining cell response to radiation by influencing DNA damage repair, we found that the “DNA repair” pathway was prevalently enriched in WT cells, showing a downregulation upon SPOP mutation (Figure 3B). Other pathways related to DNA damaging stimuli, such as “UV response”, were differentially enriched between SPOP mutant and WT cells. Specifically, “UV response down (dn)” emerged as a significant enriched gene set in F133V SPOP cells, and, consistently, the inverse pathway “UV response up” showed a coherent trend in WT SPOP cells.

To functionally characterize the role of mutant SPOP in DNA damage and repair, we evaluated the presence of nuclear γH2AX foci, a specific marker of DNA double strand breaks (DSBs), in F133V SPOP compared to WT SPOP cells. Immunofluorescence staining showed that radiation treatment induced extensive and comparable DNA damage in both WT and F133V SPOP expressing cells, as indicated by a similar number of γH2AX foci at 1 h upon irradiation (Figure 4A), confirming that the presence of mutant SPOP does not affect the induction and recognition of DNA damage upon treatment. Conversely, F133V SPOP cells showed a markedly reduced ability to remove DNA DSBs compared to WT SPOP cells, as indicated but the significantly higher number of γH2AX foci still present at 8 h following irradiation (Figure 4A).

Based on the knowledge that SPOP mutation impairs HR, we next examined the formation of nuclear RAD51 foci, a specific marker of HR, upon irradiation in WT and mutated cells. Interestingly, and consistent with the presence of γH2AX foci, RAD51 foci were markedly increased at 8 h upon irradiation in WT SPOP cells but not in F133V SPOP cells, indicating that the presence of SPOP mutation negatively interferes with the HR repair pathway (Figure 4B). Accordingly, through the assessment of the nuclear translocation of RAD51, which underlies its engagement in the HR machinery, we observed that RAD51 protein enrichment in the nuclear fraction increases in WT SPOP cells upon irradiation but not in F133V SPOP cells, further confirming that RAD51 translocation and the consequent HR machinery activation is impaired in SPOP-mutated cells (Figure 4C).

Taken together, these findings indicate that the presence of SPOP mutation, while not affecting the induction and recognition of radiation-induced DNA damage, remarkably impairs the induction of the HR DNA damage repair pathway in PCa cells by interfering with the formation of RAD51 nuclear foci.

### 2.3. Knockdown of SPOP Enhances PCa Cell Response to Ionizing Radiation

Nearly all PCa-specific SPOP mutations, including Y87N, K129E and F133V, reside in proximity of the substrate-binding cleft of its MATH domain, indicating a loss of function phenotype [2]. This evidence suggests that an altered expression of SPOP may compromise its function as well as the presence of loss-of-function mutations [6]. To investigate whether the loss of SPOP in PCa cells produces a radiosensitizing effect comparable to that observed upon SPOP mutation, we performed siRNA-based phenocopy experiments to specifically suppress endogenous SPOP in DU145 and PC-3 cells. Following siRNA-SPOP (siSPOP) transfection, SPOP transcript levels were significantly reduced compared to negative control (siNeg) in both cell lines (Figure 5A). This reduction was paralleled by a comparable decrease in the levels of SPOP protein, as shown by western blot analysis (Figure 5B), thus confirming the ability of siSPOP to efficiently suppress the transcription and expression of endogenous SPOP.

SPOP knockdown significantly inhibited DU145 and PC-3 cell proliferation over time up to 48% and 40%, respectively, compared to controls (Figure 5C) and reduced their plating efficiency in the clonogenic assay (0.13 and 0.16 in siSPOP transfected DU145 and PC-3 cells, respectively, compared to 0.22 and 0.28 of the corresponding siNeg cells). Interestingly, SPOP knockdown was also able to significantly enhance cell response to radiation in both cell lines, similarly to what was observed with SPOP mutants, suggesting that, although moderate, the slight knockdown of SPOP is sufficient to exert a significant biological effect. Specifically, the DER values were 1.42 and 1.32 in siSPOP-transfected DU145 and PC-3 cells, respectively, which are almost superimposable to that of SPOP mutants, indicating that the loss of SPOP indeed recapitulated the radiosensitizing effect induced by SPOP mutation (Figure 5D).

In addition, to explore the possibility of enhancing the radiation response suppressing SPOP through the use of a physiological negative regulator of the gene, we focused on miR-145-5p, which recently emerged as an upstream regulator of SPOP [20] and was also shown to increase the radiosensitivity of PCa models by modulating DNA repair [21]. To this purpose, DU145 cells were transfected with a synthetic miR-145 mimic, resulting in a significant downregulation of SPOP at both transcript and protein levels, although lower than that observed with siSPOP (Figure 5E,F). This evidence is in line with the miRNA characteristic of being fine tuners of gene expression. Moreover, consistent with its negative modulation of SPOP expression, miR-145 showed an inhibitory effect on cell proliferation and plating efficiency (0.17 in miR-145 transfected cells compared to 0.22 of Neg cells), although at a lower extent compared to what was observed upon siSPOP transfection (Figure 5G). Interestingly, miR-145 was able to remarkably increase DU145 sensitivity to radiation, as shown by the decreased cell surviving fractions and corroborated by the DER value of 1.49, which is even slightly greater than that observed upon siSPOP transfection (Figure 5H).

Collectively, these results substantiate our hypothesis that SPOP suppression, through the use of either siRNA or miRNA molecules, recapitulates SPOP mutation potential in influencing PCa cell response to radiation.

### 2.4. SPOP Knockdown Impairs Homologous Recombination via RAD51 and CHK1 Downregulation

To gain further insight into the mechanisms underlying the observed HR impairment in SPOP deficient cells, and based on previous evidence that SPOP interferes with the transcription of main HR mediators [14], we assessed the effect of SPOP knockdown on the expression of RAD51 and CHK1 in the two PCa cell models. siRNA-mediated SPOP suppression significantly decreased the expression of RAD51 and CHK1 mRNAs in DU145 and PC-3 cell lines, confirming that both factors of the HR machinery are regulated by SPOP at the transcriptional level (Figure 6A). Accordingly, the protein levels of the two mediators were considerably reduced upon SPOP downregulation, suggesting that the loss of SPOP function impairs HR-mediated DNA DSB repair through RAD51 and CHK1 downregulation (Figure 6B). Consistent with these findings, when the formation of RAD51 foci following irradiation was assessed in the presence or absence of SPOP knockdown, we found that RAD51 foci rapidly accumulate and persist in siNeg cells, whereas their formation was significantly reduced in siSPOP transfected cells (Figure 6C).

These results indicate that SPOP suppression leads to the downregulation of RAD51 and CHK1 and, consequently, to the impairment of the HR mechanism of DNA damage repair, hence entailing an inadequate resolution of radiation-induced DNA DSBs.

## 3. Discussion

This is the first comprehensive study aimed at comparatively assessing the potential of deregulated SPOP, via gene mutation or siRNA- and miRNA-mediated knockdown, as a radiosensitizing factor in PCa experimental models. Specifically, we found that DU145 and PC-3 cells ectopically expressing a PCa-associated SPOP mutation affecting the MATH domain (Y87N, K129E or F133V) were characterized by markedly enhanced in vitro radiosensitivity profiles, as indicated by the significantly decreased clonogenic cell survival observed at all the tested doses. In addition, an original finding emerging from the study is that all the assessed SPOP mutations showed a comparable radiosensitizing effect in the context of a specific cell line. Most importantly, we demonstrated for the first time that stable expression of F133V SPOP, which represents the SPOP mutation most frequently observed in clinical PCa [2], was able to improve the in vivo response to 5 Gy irradiation in PCa mouse xenografts.

Phenocopy experiments carried out to assess the radiation response of DU145 cells upon SPOP silencing with a specific siRNA showed that gene knockdown was able to induce a radiosensitizing effect comparable to that observed in the presence of mutant SPOP. Superimposable results were also obtained through the ectopic expression of miR-145-5p, which recently emerged as a physiological negative regulator of SPOP from a pathway analysis in prostate organoids from conditional SPOP-F133V transgenic mouse [13]. Indeed, the 3′-UTR regions of both mouse and human SPOP transcripts harbor a conserved putative miR-145 binding site [20]. Consistent with our findings, miR-145 reconstitution was previously shown to increase the radiosensitivity of PCa models [21].

Taken together, the data support the notion that SPOP downregulation, as accomplished by either gene mutation or RNAi-mediated knockdown, is consistently able to improve the radiation response of PCa models.

In accordance with the previously reported involvement of SPOP in determining cell response to radiation by influencing DNA damage repair [15,16], the comparative transcriptomic analysis performed in DU145 cells carrying WT or F133V SPOP to assess relevant pathways differentially expressed showed that the “DNA repair” gene set, which comprises all the pathways related to DNA repair mechanisms, was downregulated upon SPOP mutation. Consistently, the functional characterization of the effect of mutant SPOP in radiation-induced DNA damage and repair through the assessment of the kinetics of accumulation and removal of γH2AX foci, as a specific marker of the presence of DNA DSBs, indicated that, although treatment induced an extensive and comparable amount of DNA lesions in WT and F133V SPOP cells, the resolution of γH2AX foci was markedly delayed in mutant SPOP expressing cells. Consistent with a different presence of γH2AX foci, an increased nuclear translocation of RAD51 protein, which underlies its engagement in the HR machinery, together with a markedly enhanced presence of RAD51 foci upon irradiation was observed in WT SPOP cells but not in F133V SPOP cells, thus corroborating previous evidence indicating that SPOP mutation negatively interferes with the HR repair pathway [7]. Superimposable results were obtained in DU145 and PC-3 cells upon RNAi-mediated SPOP knockdown. Moreover, in accordance with recently reported findings showing that SPOP promotes the transcriptional expression of DNA repair factors including RAD51 and CHK1 and that SPOP knockdown impairs RAD51 foci formation and CHK1 activation at the transcriptional level in response to replication stress [14], we found that the expression of the two genes was markedly reduced at both RNA and protein levels in siSPOP-transfected PCa cells.

Overall, these findings indicate that the presence of mutant or downregulated SPOP, while not affecting the induction and recognition of radiation-induced DNA damage, significantly impairs the HR repair pathway in PCa cells by interfering with the formation of RAD51 nuclear foci in response to DNA DSB induction. In addition, results from the study corroborate the notion that mutant SPOP could represent a novel biomarker of radiation sensitivity and also highlight the possibility to develop novel strategies of radiosensitization based on the use of SPOP downregulation approaches.

## 4. Materials and Methods

### 4.1. Experimental Models

The human DU145 and PC-3 PCa cell lines (American Type Culture Collection, ATCC, Manassas, VA, USA) were cultured in RPMI-1640 medium supplemented with 10% fetal bovine serum (FBS). Cell lines were authenticated and monitored via genetic profiling using short tandem repeat analysis (AmpFISTR Identifiler PCR amplification kit, Thermo Fisher Scientific Inc., Waltham, MA, USA).

### 4.2. Cell Transfection

For the ectopic expression of WT and mutant SPOP, cells were seeded at the density of 8000 cells/cm^2^ in culture vessels and transfected with 50 μg pMCV6-Entry vectors expressing WT or mutant (Y87N, F102C, F133V) SPOP protein tagged with a C-terminal Myc/FLAG epitope (kindly provided by Drs. M.A. Rubin and C. Barbieri [2]), using Lipofectamine 2000 (Thermo Fisher Scientific Inc.). For in vivo experiments, DU145 clones stably expressing WT SPOP and F133V SPOP were established. Cells were transfected according to Lipofectamine 3000 protocol (Thermo Fisher Scientific Inc.). Briefly, cells were seeded in 6-well plates (5 × 10^5^ cells/well) and 24 h later transfected with vectors (10 pg). After 72 h, cells were selected using 0.4 mg/mL zeocin for 5 weeks. Stable transfectants were stored in liquid nitrogen at −196 °C. When cultured, transfected cells were maintained in the presence of 0.2 mg/mL of zeocin.

To downregulate endogenous SPOP expression, cells were transfected for 4 h with 20 nM mirVana miRNA mimic (miR-145 MC11480, Thermo Fisher Scientific Inc., Waltham, MA, USA) and mirVana miRNA mimic Negative control (miR-Neg Negative control #1, Thermo Fisher Scientific Inc.) using Lipofectamine RNAiMAX (Thermo Fisher Scientific Inc.), or with 20 nM Silencer Select Pre-Designed siRNA-SPOP (siSPOP, Silencer Select Pre-Designed siRNA-SPOP_s15955, Thermo Fisher Inc.) and Silencer Select Negative Control (siNeg) using Lipofectamine 2000 (Thermo Fisher Scientific Inc.), according to the manufacturer’s instructions. Specifically, miR-145-based experiments were carried out on DU145 since we already have a lot of experience in reconstituting different miRNAs in this cell line [22,23] and have already obtained reproducible results.

### 4.3. Cell Proliferation

Cells were seeded in 6-well plates at the density of 4000 cells/cm^2^ and transfected with miRNA mimics or siRNAs 24 h later. Cells were detached and counted through a Coulter Counter (Beckman Coulter Life Sciences, Brea, CA, USA) at different time points (24, 48, 72 and 96 h). Data were reported as number of proliferative cells ×10^3^.

### 4.4. Clonogenic Assay

Transfected cells were irradiated with increasing doses (2–8 Gy) delivered as a single dose using the 137Cs γ-irradiator IBL-437 (dose rate 5.2 Gy/min). Specifically, cell samples were placed in the central region of the irradiation cavity where the dose uniformity is ensured with a tolerance of 5%. The time of irradiation takes into account the correction for radioisotopic decay, which is monitored every 6 months. Irradiated cells were seeded in triplicate at increasing densities (500–8000 cells/well), in 6-well plates cultured in RPMI complete medium. After 10 and 14 days, DU145 and PC-3 colonies, respectively, were fixed with 70% ethanol, stained with crystal violet in 70% ethanol, and counted (only colonies consisting of at least 50 cells were considered). The plating efficiency was calculated as the ratio of the number of colonies to the number of seeded cells. The surviving fraction was calculated as the ratio of the plating efficiency of the irradiated cells to that of the non-irradiated ones. The dose enhancement ratio (DER) was calculated as the ratio of the radiation dose required to obtain a surviving fraction of 0.1 in control cells to that required to obtain the same SF in mutant/silenced SPOP cells. GraphPad Prism was used to fit the linear-quadratic model of cell clonogenic survival based on the equation Y = exp (−1 × (A × X + B × X^2^)), which has been established to be a suitable model for the description of cell death upon irradiation [24].

### 4.5. In Vivo Experiments

All animal experiments were approved by the Ethics Committee for Animal Experimentation of Fondazione IRCCS Istituto Nazionale dei Tumori and by the Italian Health Ministry (#1120/2015PR). For the generation of xenografts, DU145 cell line was used since it is characterized by an efficient engraftment in mice. Ten million DU145 cells transfected with WT and F133V SPOP vectors were injected into the right flank of eight-week-old male SCID mice. When tumors reached ~300 mm^3^ (Width^2^ × Length/2), mice were randomly assigned to mock or radiation treatment groups (*n* = 8). Using a micro-CT/microirradiator (225Cx, Precision X-ray) mice were exposed to 5 Gy single dose irradiation, a dose that emerged as the best compromise between efficacy and safety, based on our previous experience [22,23] and literature data [25,26].

Specifically, mice were anesthetized with a solution of ketamine (100 mg/kg) + xylazine (5 mg/kg) and imaged through cone-beam computer tomography (CBCT) using a micro-CT/microirradiator (225Cx, Precision X-ray) with filtration of 2 mm of aluminum. The resulting imaging scan was used for the delineation of tumor contouring using SmART Plan software.

For mice treatment, two parallel opposed fields were created to cover the target with the prescribed dose, delivering irradiation through 0.3 mm of copper filtration and a square collimator (1 × 1 cm). After the Monte Carlo-based dose calculation, the dose-volume histogram (DVH) was checked to ensure that 100% of the target received 100% of the prescribed dose (the gross tumor volume was contoured). Sparing of animal body and other organs were ensured by using tangential radiation beams. Finally, to expose all mice to the same conditions, radiotherapy was delivered in a single fraction of 0 Gy and 5 Gy for the control group and the treatment group, respectively. To determine tumor growth, a Vernier caliper was used to regularly measure tumor size.

### 4.6. Total RNA Extraction and RT-qPCR

RNA extraction and cDNA synthesis were performed to assess miRNA and gene expression levels. RNA was isolated using QIAzol Lysis Reagent and a miRNeasy Mini Kit (QIAGEN, Hilden, Germany) according to the manufacturer’s instructions. cDNA was synthesized using a miScript II RT Kit (Thermo Fisher Scientific Inc.).

Quantification of gene or miRNA expression was assessed using RT-qPCR with the following TaqMan microRNA or gene expression assays (Thermo Fisher Scientific Inc.): SPOP (Hs00737433_m1), RAD51 (Hs00947967_m1) and CHEK1 (Hs00967506_m1).

For comparative analyses, GAPDH (TaqMan Gene Expression Assay, Hs02786624_g1) and SNORD (TaqMan non coding RNA assay, Hs04931161_g1) were used as endogenous controls for genes and miRNA, respectively. RT-qPCR results were reported as relative quantity (RQ = 2-ddCt) with respect to a calibrator sample using the comparative Ct (ddCt) method.

### 4.7. Immunoblotting Analyses

Cell lysates (20 μg) were fractioned using SDS-PAGE and transferred onto nitrocellulose membranes using standard protocols. Membranes were blocked in PBS-Tween-20/0.5% skim milk and probed overnight with the following antibodies: RAD51 (MA5-14419, Invitrogen, Carlsbad, California, USA), gamma H2AX (phospho S139) (ab11174, Abcam, Cambridge, UK), H2AX (ab11175, Abcam), SPOP (ab137537, Abcam), CHK1 (ab47444, Abcam). GAPDH (G8795, Sigma-Aldrich, St. Louis, Missouri, USA), β–tubulin (T521, Sigma-Aldrich), Vinculin (V9131, Sigma-Aldrich) and HDAC (877-616-cell, Cell Signaling, Danvers, MA, USA) were used as equal protein loading controls.

Detection of bound peroxidase linked secondary antibodies was assessed using a Novex ECL, HRP Chemiluminescent Substrate Reagent Kit (Thermo Fisher Scientific Inc.). Original whole blots were cropped to generate figure panels showing relevant proteins. Cropped images were subjected to uniform image enhancement of contrast and brightness. Molecular weights were determined using the colorimetric Precision Plus Protein Standard (Bio-Rad, Hercules, CA, USA). Whole blots showing all the bands with molecular weight markers were provided in Appendix A.

### 4.8. Immunofluorescence

Transfected cells grown on SlideFlasks (Nunc Lab-Tek Flask on Slide, 17092, Thermo Fisher Scientific Inc.) were fixed with 4% formaldehyde and permeabilized with cold methanol:acetone, 1:1 solution. Cells were probed with primary antibodies for phospho-Histone H2AX (ab11174, Abcam), RAD51 (MA5-14419, Invitrogen) and subsequently with Alexa Fluor488-labeled (Thermo Fisher Scientific Inc.) secondary antibodies for 1 h at room temperature. Based on clonogenic cell survival results, we chose 6 Gy as an informative dose to assess phospho-H2AX foci. Nuclei were counterstained with DAPI (Thermo Fisher Scientific Inc.). Images were acquired using a Nikon Eclipse E600 microscope with ACT-1 software (Nikon, New York, NY, USA). At least ten single-layer fluorescence images were taken per condition, in triplicate, and a total of >300 cells/condition were manually counted. Based on a conventional threshold reported in literature [27,28], only cells containing >10 foci were considered positive.

### 4.9. Bioinformatic Analyses

Transcriptomic data from DU145 prostate cells ectopically expressing WT SPOP (three independent samples) and F133V mutant SPOP (three independent samples) was generated in our laboratory using Illumina HumanHT-12 v4 arrays. The genomic alteration and the wild-type control cells were conducted in triplicate for bioinformatics analysis. Raw data were log2-transformed and normalized using the robust spline method implemented in the lumi package [29]. Normalized data were filtered removing probes with neither at least one detection *p*-value  <  0.01 across samples nor an associated official gene symbol; for probes mapping on the same gene symbol, the one with highest variance was selected.

Gene expression data was deposited at Gene Expression Omnibus, with accession number GSE147090. Differential expression was estimated in terms of *t*-value, using the limma Bioconductor package [30]. Significance was provided in terms of a false-discovery rate (FDR) to take into account the adjustment for multiple hypotheses testing. A *t*-value preranked gene set enrichment analysis (GSEA v4.0.3) [31] was performed using Hallmark gene sets and GO biological process collection v7 of the Molecular Signature database (MSigDB) [32]. An FDR *q*-value threshold of 0.05 was used to assess a significant enrichment.

All these analyses were conducted in R environment.

### 4.10. Statistical Analyses

Data are shown as mean values ± SD from at least three independent experiments. Statistical analysis was performed using two-tailed Student’s *t* test. *p*-values < 0.05 were considered statistically significant.

## 5. Conclusions

In the precision medicine era, a reliable definition of the role of mutant SPOP as a radiosensitizing factor could contribute to tailoring therapy of individual PCa patients by delivering a genomic-adjusted radiation dose, instead of a uniform dose (i.e., one-size-fits-all). In addition, the possibility to enhance the radiation response through siRNA/miRNA-mediated SPOP downregulation will open an avenue for the design of radiosensitizing approaches through the targeting of gene negative regulators.

## Figures and Tables

**Figure 1 cancers-12-01462-f001:**
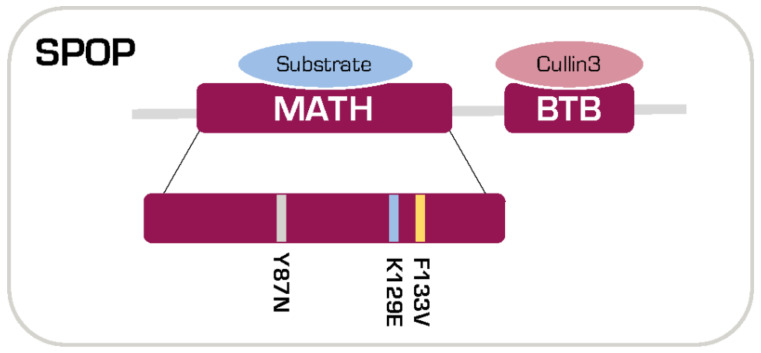
Illustration of SPOP domains and localization of prostate cancer-specific mutations.

**Figure 2 cancers-12-01462-f002:**
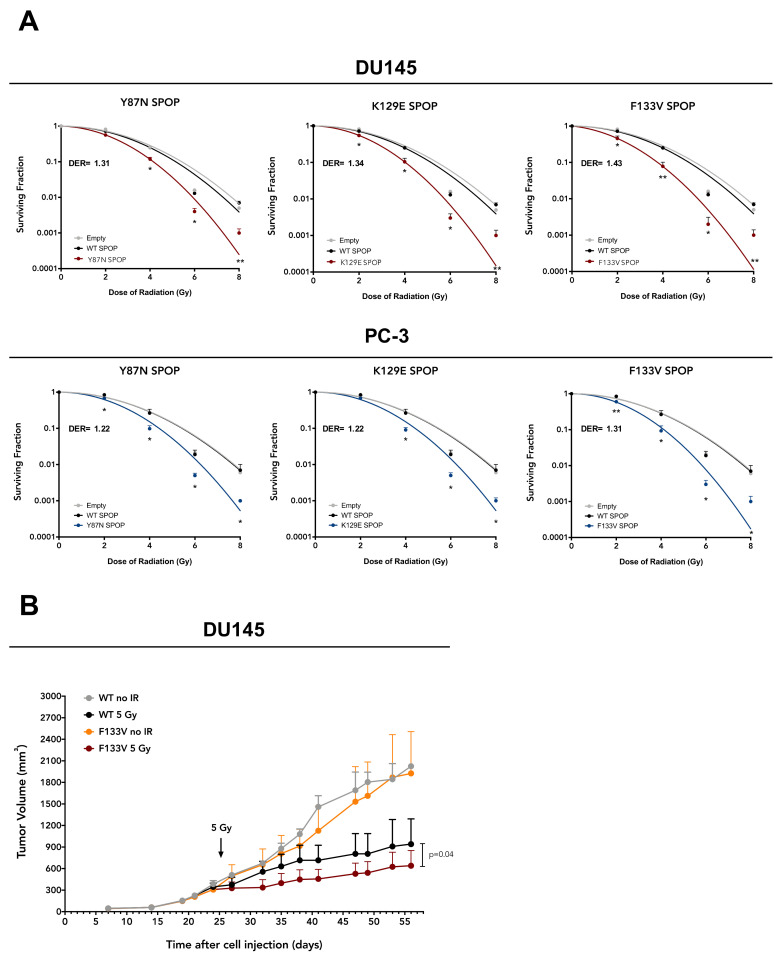
SPOP mutations sensitize PCa cell and animal models to ionizing radiation. (**A**) Clonogenic cell survival curves of DU145 and PC-3 cells ectopically expressing WT or mutant (Y87N, K129E, F133V) SPOP and exposed to increasing doses of irradiation (2, 4, 6 and 8 Gy). The surviving fraction is reported as mean ± SD values from three independent experiments. The level of significance was represented as * *p* < 0.05, ** *p* < 0.01, Student’s *t*-test. The dose enhancement ratio (DER) was calculated as the dose (Gy) for the radiation plus mutant SPOP divided by the dose (Gy) for radiation plus WT SPOP at a surviving fraction of 0.1. (**B**) DU145 cells stably expressing WT or F133V SPOP (1 × 10^7^) were subcutaneously implanted into SCID mice. When tumors reached ~300 mm^3^, mice were randomly assigned to four groups (8 mice/group) and treated with 5 Gy single dose irradiation locally delivered to the tumor. Tumor growth volume (mm^3^) was measured with a Vernier caliper on indicated days after cell injection.

**Figure 3 cancers-12-01462-f003:**
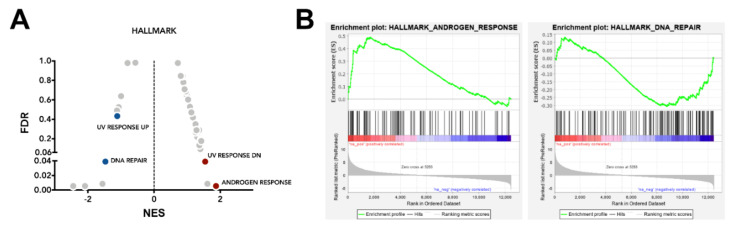
F133V SPOP mutation deregulates DNA repair-related pathways. (**A**) Volcano plot of Hallmark collection gene sets from Molecular Signature database (MsigDB); normalized enrichment score (NES) and false discovery rate (FDR, 0.05 threshold to assess significance) were plotted. Dots highlight the enriched gene sets in F133V SPOP mutated (red) and in WT (blue) cells. (**B**) Gene set enrichment analysis (GSEA) enrichment plot of “androgen response” and “DNA repair” in the comparison between F133V SPOP and WT DU145 cells.

**Figure 4 cancers-12-01462-f004:**
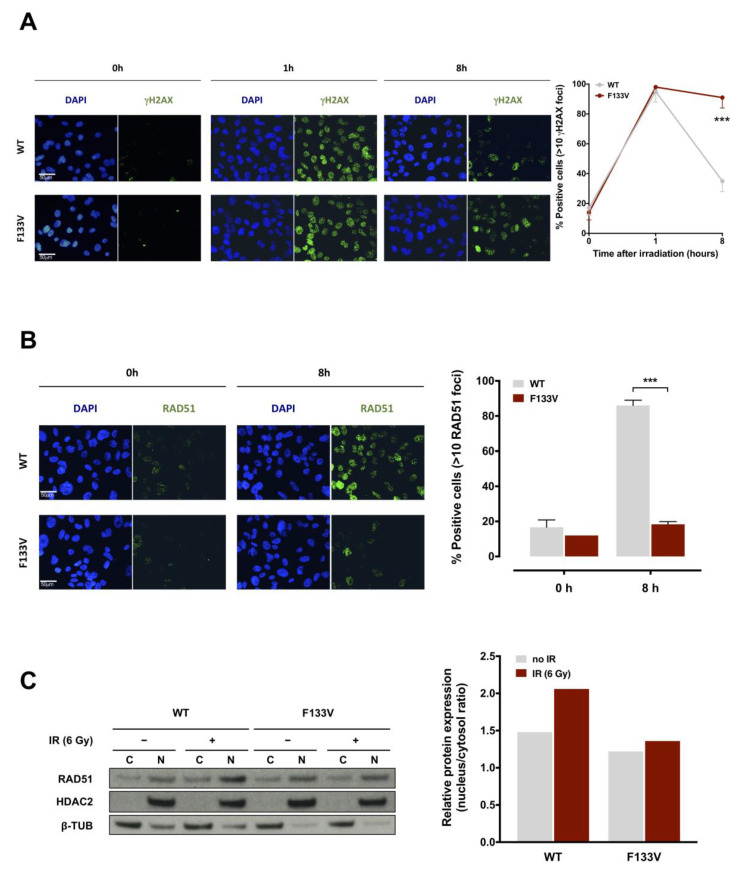
F133V SPOP impairs cell ability to recover from radiation-induced DNA damage. (**A**) (Left panel) Representative immunofluorescence microphotographs of nuclear γH2AX foci in DU145 WT or F133V SPOP cells at 0, 1 and 8 h upon 6 Gy irradiation. (Right panel) Quantification of γH2AX foci in WT or F133V SPOP DU145 cells, expressed as mean percentage of cells containing >10 γH2AX foci at 0, 1 and 8 h upon irradiation. Data are reported as mean ± SD values. The level of significance was represented as *** *p* < 0.001, Student’s *t*-test. (**B**) (Left panel) Representative immunofluorescence microphotographs of RAD51 nuclear foci in DU145 WT or F133V SPOP cells at 0 and 8 h upon irradiation (6 Gy). (Right panel) Quantification of RAD51 foci expressed as mean percentage of cells containing >10 RAD51 foci at 0 and 8 h upon irradiation (IR). Data are reported as mean ± SD. (**C**) Western blot analysis and relative quantification of RAD51 protein levels in nucleus/cytosol fractions of DU145 WT or F133V cells at 0 and 8 h upon irradiation (6 Gy). HDAC2 and β-tubulin were used as equal protein loading controls for nuclear and cytosolic fractions, respectively. Relative protein expression was calculated as nuclear/cytosolic ratio of normalized RAD51 levels.

**Figure 5 cancers-12-01462-f005:**
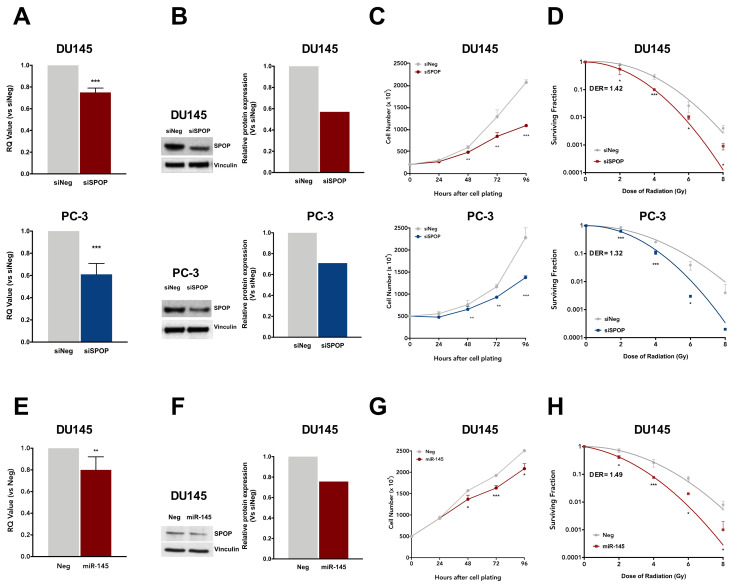
SPOP knockdown, through siRNA-SPOP (siSPOP) or miR-145 transfection, enhances cell response to radiation. (**A**) qRT-PCR detection of SPOP transcript levels in DU145 (upper panel) or PC-3 (lower panel) at 48 h upon transfection with siSPOP, compared to control cells, normalized to GAPDH. Data are reported as relative quantity (RQ) ± SD with respect to siNeg transfectants. (**B**) Western blot analysis and relative quantification of SPOP protein levels in DU145 and PC-3 cells at 48 h upon siSPOP transfection. Vinculin was used as endogenous control. (**C**) Cell proliferation curves of siNeg and siSPOP at 24, 48, 72 and 96 h upon transfection. Data are indicated as number of cells ×10^3^ and are reported as mean ± SD values (*n* = 3). (**D**) Clonogenic cell survival of DU145 and PC-3 cells upon transfection with siNeg or siSPOP. The surviving fractions are reported as mean ± SD values from three independent experiments. The dose enhancement ratio (DER) was calculated as the dose (Gy) for the radiation plus siSPOP divided by the dose (Gy) for radiation plus siNeg at a surviving fraction of 0.1. (**E**) qRT-PCR detection of SPOP transcript levels in DU145 cells at 48 h upon transfection with miR-145, compared to control cells, normalized to GAPDH. Data are reported as relative quantity (RQ) ± SD with respect to Neg cells. (**F**) Western blot analysis and relative quantification of SPOP protein levels in DU145 cells at 48 h upon miR-145 transfection. Vinculin was used as control. (**G**) Cell proliferation curves of Neg and miR-145 at 24, 48, 72 and 96 h upon transfection. Data are indicated as number of cells ×10^3^ and are reported as mean ± SD values from three independent experiments. (**H**) Clonogenic cell survival of Neg or miR-145-transfected DU145 cells. The surviving fractions are reported as mean ± SD values from three independent experiments. The dose enhancement ratio (DER) was calculated as described above. The level of significance was represented as * *p* < 0.05, ** *p* < 0.01, *** *p* < 0.001, Student’s *t*-test.

**Figure 6 cancers-12-01462-f006:**
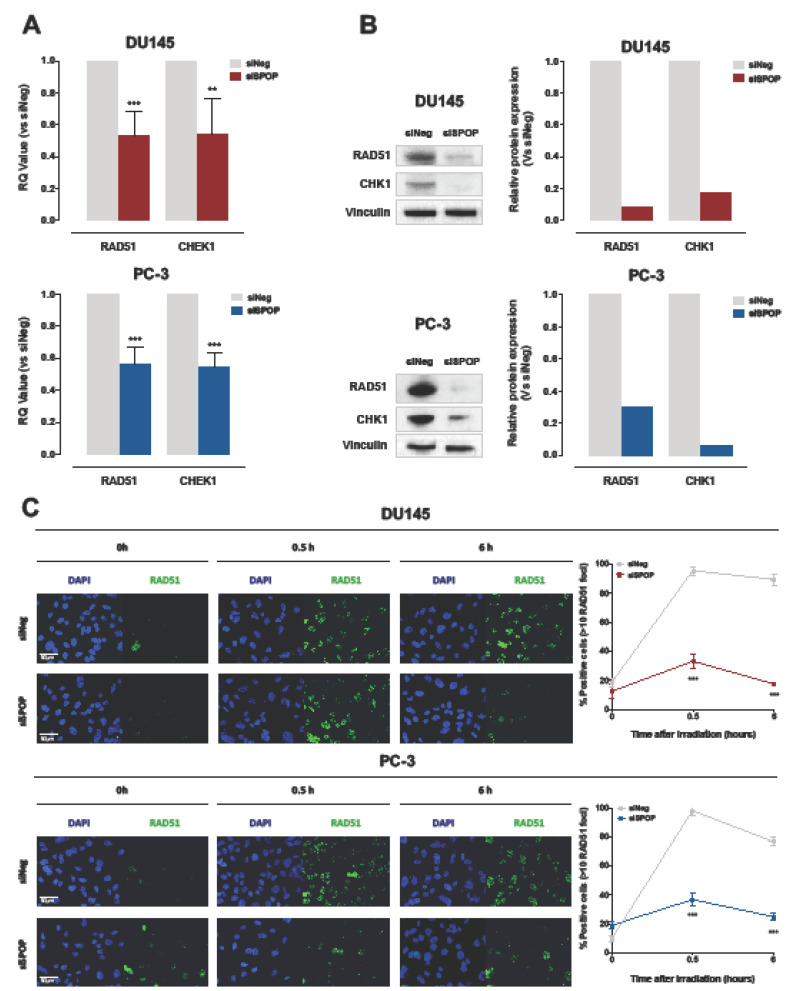
SPOP knockdown impairs HR via RAD51 and CHK1 downregulation. (**A**) qRT-PCR detection of RAD51 and CHEK1 transcript levels in DU145 (upper panel) or PC-3 (lower panel) cells at 48 h upon siSPOP transfection, compared to controls, normalized to GAPDH. Data are reported as relative quantity (RQ) ± SD with respect to siNeg transfectants. (**B**) Western blot analysis and relative quantification of RAD51 and CHK1 protein levels in DU145 and PC-3 cells at 48 h upon siSPOP transfection. Vinculin was used as equal protein loading control. (**C**) Representative immunofluorescence microphotographs of nuclear RAD51 foci (cell nuclei: blue; RAD51 foci: green) in DU145 (upper panel) or PC-3 (lower panel) cells at 48 h upon transfection with siSPOP at 0, 0.5 and 6 h after exposure to 6 Gy irradiation and relative quantification, expressed as mean percentage of cells containing >10 RAD51 foci at 0, 0.5 and 6 h after exposure to 6 Gy irradiation. Data are reported as mean ± SD values from three independent experiments. The level of significance was represented as ** *p* < 0.01, *** *p* < 0.001, Student’s *t*-test.

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
