# Peer review of "SPOP Deregulation Improves the Radiation Response of Prostate Cancer Models by Impairing DNA Damage Repair"

_cancers, 2020, doi:10.3390/cancers12061462_

Round 1

Reviewer 1 Report

Manuscript “SPOP deregulation improves the radiation response of prostate cancer models by impairing DNA damage repair” by Rihan El Bezawy et al demonstrates how SPOP protein mutations or mRNA downregulation increases sensitivity to radiation in DU145 and PC3 prostate cancer cell lines. Authors also shown evidences that SPOP silencing also resulted in the impairment of Homologous Recombination (HR). Overall, methods used are adequate, results are correctly presented and conclusions are ok. Before acceptance, some points should be addressed.

  1. SPOP-mutated prostate cancers possess the highest transcriptional activity of androgen receptor. However authors have analyzed the effect of SPOP mutation/underexpression in prostate cancer cell lines DU145 and PC-3 that are hormone-insensitive. Would the results be the same in prostate cancer cells that respond to androgens? Would androgen-sensitive prostate cancer cells with SPOP mutation/underexpression be more radiosensitive than wild type SPOP? Would they display aberrant HR and DNA damage response (DDR), with and without wild-type SPOP?
  2. Fig. 2B. Authors report that F133V SPOP DU145 cells display enrichment of androgen-response signatures. Would the enrichment be associated with response to antiandrogens treatment in these F133V SPOP DU145 cells? Would both radio and antiandrogen therapies synergically act onto these cells?
  3. Would RAD51 and CHEK1 be modulated by SPOP also at the protein level? Would mutant SPOP induce protein degradation on them, besides transcriptional regulation?

Reviewer 2 Report

I have read with interest the manuscript "SPOP deregulation improves the radiation response of prostate cancer models by impairing DNA damage repair" by El Bezawy et al.

This is a consize and comprehensive study of SPOP, a Cullin 3 adaptor in sensitation to ionizing radiation, DNA damage response and regulation of homologous recombination through effects on expression of RAD51 and CHEK1.

I am in general impressed by the clarity of the study but have a few comments that might, if considered, improve the manuscript:

  1. For easier reading of abstract and introduction it would be helpful when describing SPOP to have an illustration of the protein and its domains as well as indications of the mutations mentioned.
  2. In introduction spell out or describe what PTEN means for those not directly in the field.
  3. Figure 1 The stars indicating specificity are too small, meaning impossible to see. The same is true for fig 4. Also, please indicate what "noRT" means.
  4. For all statistics - is the two-tailed students t test the best statistical method in all cases?
  5. In figure 4 The knock down effect of siSPOP is to say the least very modest, both at RNA and protein levels. This should at least be commented on, which is not the case as far as I can see.
  6. Also in Fig 4. The quantification of the Western blots are without error bars. Does this mean that there is no variation? or that this quantification is only valid for a representative WB (the one shown)? or that the Western blots were only done once?
  7. In the Discussion one of the two "Taken together" lines 331 and 352 could be varied.

Reviewer 3 Report

Overall, this was a sound and well thought out study investigating the effect of SPOP on radiosensitization of prostate cancer cells. Here are my comments: 

1) The authors mentioned that plating efficiency was reduced for cells transfected with SPOP mutant. It might be beneficial to address how much of the DER is due to reduction in plate efficiency for each construct vs. the effect of radiation itself. 

2) The authors claim that "ectopic expression of F133V SPOP enhanced the effect of radiation also in vivo, as indicated by a statistically significant reduction in tumor growth upon irradiation compared to SPOP xenografts". However, this claim doesn't seem to be supported by Figure 1B given how wide the error bars are. 

3) The transcriptome analysis has the potential to strengthen the authors' claim that the presence of SPOP mutation deregulates DNA repair pathway, but there was no validation of expression of the differentially expressed genes that came up during the analysis. Doing so would only strengthen their argument.  

4) There is a typo on line 57 (need to correct "transriptome")

Reviewer 4 Report

Cancers-783382

In this study, the authors determined the effects of SPOP on radiosensitivity of prostate cancer cells and xenograph mice. They found that knockdown of SPOP and overexpression of SPOP mutants significantly increased the radiosensitivity of prostate cancer cells. They further demonstrated that silencing of SPOP reduced the formation of RAD51 foci and expression of RAD51 and CHK1 in prostate cancer cells. The results indicate that downregulation of SPOP increased radiosensitivity of prostate cancer cells by reducing the homologous recombination. The study is interesting. However, considering several previous studies have already shown that SPOP loss results in the reduction of RAD51 formation in responding to radiation-induced double strand DNA breaks, it is not clear what new knowledge this study can contribute to the field. This needs to be addressed in the introduction of the manuscript. Also, the following specific points need to be addressed.

  1. Control experiments showing the effects of overexpression of wild-type SPOP on the radiation sensitivity of the prostate cancer cells are needed.
  2. Figure 3C, why HDAC2 was detected here? Also, the gel loading control, beta-TUB need be improved. This can help to show more evident difference between C and N.
  3. Figure 4E and 4F, it is hard to see the significant difference between Neg and miR-145. However, it is surprising that the difference exhibits statistical significance in 4G and 4H. This needs to be discussed.
  4. Why CHK1 expression was also reduced by SPOP?
  5. It is interesting that Figure 2 showed that UV response and DNA repair pathways was affected by the overexpression of SPOP mutant F133V. What are these pathways? Are there other DNA repair pathways? More information and discussion about these need to be provided.

Reviewer 5 Report

This paper aims to define the role of SPOP deregulation on prostate cancer radiosensitivity using cell line models, DU145 and PC3. This is an interesting and well written manuscript but there are a number of issues to be resolved, mainly involving clarification and justification of experimental design. 

The following points should be included in the revised manuscript:

In the in vivo experiment why were DU145 cells used and not PC3 cells?

In the in vitro experiment, doses up to 8 Gy were used. Please give the rationale for choosing the 5 Gy dose point for the in vivo experiment.

For the measurement of gammaH2AX foci, please give the rationale as to why an eight hour timepoint was chosen to look for residual damage?  Also why was a dose of 6 Gy used? Please give the rationale for scoring more than 10 foci and include more detail on this in the Methods section 4.8.

In figure 3A, are the results significant at eight hours?

In figure 3B, why wasn't a one hour time time point used in addition to an eight hour timepoint?

In figure 3C, why are there no error bars?

For the transfection with miR 145, why were only DU-145 cells used?

In figure 4B and F why are there no error bars?

In figure 4C and G, please keep the same y axes and perhaps change to cell number since profileration or viability were not directly measured, only cell number.

In figure 5B, why are there no error bars?

In figure 5C why was a 0.5 hour time point used instead of one hour timepoint like in Figure 4?

In the Methods section, please give a description of how the irradiation was carried out including dosimetry.

In Methods 4.2, 4.3, please check superscripts on units.

Round 2

Reviewer 1 Report

Authors answer adequately the points I raised. Manuscript is OK for publication.

Author Response

We thank the reviewer for the positive comments.

Reviewer 4 Report

The responses that addressed point 4 and 5 need to be included in the Discussion.

Author Response

REVIEWER 4

The responses that addressed point 4 and 5 need to be included in the Discussion.

Response to reviewers’ comments

As requested by the reviewer, comments addressing points 4 and 5 have been included within the discussion at lines 574-589 and 561-562 respectively.

Reviewer 5 Report

The authors have addressed all of this Reviewer's concerns in the coverletter. However, some of the points regarding justification of the experimental design still need to be included in the manuscript itself.

Please add brief comments (based on the responses in the coverletter) to the Methods section concerning points 1, 2, 3 (please add refs to 'Based on a conventional 470 threshold reported in literature, only cells containing >10 foci were considered positive.') and 7.

Author Response

REVIEWER 5

The authors have addressed all of this Reviewer's concerns in the coverletter. However, some of the points regarding justification of the experimental design still need to be included in the manuscript itself.

Please add brief comments (based on the responses in the coverletter) to the Methods section concerning points 1, 2, 3 (please add refs to 'Based on a conventional 470 threshold reported in literature, only cells containing >10 foci were considered positive.') and 7.

Response to reviewers’ comments

As suggested by the reviewer, comments concerning points 1,2, 3, and 7 have been added to the manuscript at lines 663-664, 667-669, 732-733, and 624-626, respectively. Moreover, references have been provided for foci threshold as requested (line 737).